# Optimal Water Backwashing Condition in Combined Water Treatment of Alumina Microfiltration and PP Beads

**DOI:** 10.3390/membranes12010092

**Published:** 2022-01-15

**Authors:** Hyungmin Cho, Gihoon Yoon, Minjae Kim, Jin Yong Park

**Affiliations:** Department of Environmental Sciences & Biotechnology, Hallym University, Chunchon 24252, Korea; tlr0987@naver.com (H.C.); dbsrlgns416@naver.com (G.Y.); vhalrj@naver.com (M.K.)

**Keywords:** microfiltration, water backwashing, photo-oxidation, polypropylene bead, combined treatment, water treatment, alumina

## Abstract

Membrane fouling is a dominant limit of the membrane separation process. In this research, the optimal water backwashing to solve the membrane fouling problem was investigated in the combined water treatment process of alumina MF and pure polypropylene (PP) beads. Additionally, the influence of membrane shape (tubular or seven channel) was examined, depending on the water backwashing period. The optimal backwashing time (BT) could be 20 s in the combined water treatment process, because of the highest total treated volume (V_T_) in our BT 6–30 s conditions. The optimal backwashing period (BP) could be 6 min, because of the minimum membrane fouling and the maximum V_T_ in the combined process of tubular alumina MF and PP beads. The resistance of reversible membrane fouling (R_rf_) showed a major resistance of total membrane fouling, and that of irreversible membrane fouling (R_if_) was a minor one, in the combined process using tubular or seven channel MF. The R_if_ showed a decreasing trend obviously, as decreasing BT from NBW to 2 min for seven channel MF. It means that the more frequent water backwashing could be more effective to control the membrane fouling, especially irreversible fouling, for seven channel membranes than tubular membranes.

## 1. Introduction

In the membrane separation process, usually, membrane fouling was accomplished by the adsorption-precipitation of organic and inorganic compounds on the membrane surface or inside the membrane and leads to decreasing the permeate flux, increasing membrane cleaning costs, and reducing the membrane life. Strategies for reducing membrane fouling remain insufficient, that is, the major hindrance in the successful application of membrane separation technology, although considerable progress has been made in membrane fouling control [1,2]. Furthermore, the fouling behavior, fouling factors and fouling control strategies were reviewed in membrane bioreactors (MBRs) [1]. The modified membranes were investigated for their anti-biofouling performance in the viewpoints of the anti-adhesion and anti-bacterial effects. The anti-adhesion method which prohibits the initial attachment of bacteria on a membrane surface was more efficient than the anti-bacterial approach that focuses on killing bacteria already involved on the membrane surface [2].

Natural organic matter (NOM) is a dominant constituent of membrane fouling in low-pressure membrane filtration. Numerous protective procedures to restrict NOM fouling have been established and comprehensively verified, such as coagulation, oxidation, ion exchange, carbon adsorption, and mineral oxide adsorption [3]. A new adsorbent, such as heated aluminum oxide powder, was employed in an absolutely automated pilot water treatment process to exclude NOM in the surface water [4]. 

To control membrane fouling, backpulsing/backwashing can be applied periodically to remove reversible fouling [5,6]. When the performance of the membrane decreases by 50–60%, chemical cleaning is needed to remove irreversible fouling and to restore the membrane performance [7]. In this research, periodic water backwashing was performed to inhibit the membrane fouling. In recent times, the combined process of membrane separation and photo-oxidation by ultraviolet (UV) irradiation can effectively solve the membrane fouling problem, mentioned above [8]. Moreover, the influence of UV irradiation on nanofiltration (NF) membrane biofouling had been researched in a pilot scale, and the two pilots were fed either granular activated carbon (GAC)-filtered water or UV-irradiated GAC-filtered water. UV pre-treatment did not give an impact on the organic carbon concentration; however, it was related to a much lower longitudinal pressure drop (LPD) increase, and moderate permeate flux decrease [9].

The combined technology not only retains the benefits of each technology but also yields synergistic properties to solve the limits of only a method. Furthermore, the contaminants like NOM can be oxidized by UV irradiation, and organic matters are reduced partially by controlling the residence time in the reacting process. In other words, the membrane separation has a limitation as a selective barrier that only molecules smaller than its pore size can be separated. In conclusion, the combined process could increase the photo-oxidation effectiveness and acquire outstanding effluent quality. Moreover, the influence of UV irradiation on the nano-hybrid PES-NanoZnO membrane in terms of flux and rejection efficiency has been discussed [10]. Additionally, an evaluation of the treatment effectiveness of surface water in a combined process of numerous advanced oxidation processes and ultrafiltration (UF) was published [11]. In this research, a combined process of alumina ceramic membrane and pure PP beads with UV irradiation was applied to advanced water treatment for the purpose of high water quality.

Ceramic membranes employed in this research usually have three times higher costs than polymeric membranes with similar membrane surfaces; however, those have various advantages, which are mechanical, thermal and chemical resistance, and a long lifetime. Those were economical, compared with polymeric membranes, because of higher permeate flux and practically permanent lifetime [12]. Currently, the modified and improved ceramic membranes have been applied extensively in water or wastewater treatment in the world. The various methods applied for ceramic membrane modification, focusing on utilization in oily wastewater were compared [13]. Ceramic membrane application in the treatment of emulsion wastewater was the development trend, and their useful guidance and reference were provided [14]. The impact of the characteristics of soluble algal organic matter (AOM) on the membrane fouling was examined for seven-channel tubular ceramic microfiltration (MF) membranes at a lab scale [15]. The effect of the interaction between aquatic humic substances and the AOM resulting from Microcystis aeruginosa was investigated on the membrane fouling of a ceramic MF [16].

Photo-oxidation has a lot of advantages, which are high effectiveness, low energy consumption and a wide range of applications. The mechanism of the photo-oxidation process is to mineralize organic compounds to small inorganic molecules by the oxidization of most of them, specifically non-biodegradable organic pollutants. Furthermore, it is one of the excellent processes of advanced water treatment. For these motivations, the photo-oxidation process, that was employed in this research, has been applied widely [17,18,19,20,21]. Electro-oxidation of aniline applying boron-doped diamond (BDD) electrodes was more effective than UV/H_2_O_2_ because it has an 87% lower operational cost [17]. The photo-oxidation that happened in the water column could be utilized to progress the destruction of residual organic matter in the solution by sustaining the plasma after processing a given amount of organic liquids [19]. Atomic Force Microscopy images showed smoother surfaces after UV photo-oxidation; however, atom treatment resulted in trivial changes in surface roughness. Rinsing the treated surfaces with ethanol solvent partially decreased it, representing the formation of a weak boundary during treatment [20]. The photo-oxidation of clopyralid included either hydroxylation or dechlorination of the ring; however, metaldehyde underwent hydroxylation and produced acetic acid as a major end product [21]. In addition, degradation of humic acid (HA), which was contained in a synthetic solution used in this research, via photoelectrocatalysis (PEC) process and corresponding disinfection byproduct formation potential (DBPFP) were investigated, and the PEC process was found to be effective in reducing dissolved organic carbon concentration [22].

In our research group, the results for the effect of water backwashing and PP beads in the combined water treatment process of various ceramic membranes and titanium dioxide (TiO_2_) photocatalyst-coated polypropylene (PP) were published in Desal. Water Treat. [23,24]. On the other hand, roles of adsorption and photo-oxidation in the combined water treatment process of tubular carbon fiber UF and pure PP beads with UV irradiation and water backwashing was reported by our group in Desal. Water Treat. [25]. The final dimensionless permeate flux (J_180_/J_0_) after 180 min’s operation increased as PP beads increased in the combined water treatment of carbon fiber UF membrane and PP beads [26]. Therefore, the PP beads could increase the membrane lifetime in this combined water process.

In this research, the optimal water backwashing was investigated on membrane fouling and treatment efficiency of dissolved organic matter (DOM) and turbidity in the combined water treatment process of seven channels or tubular alumina MF membranes and pure polypropylene (PP) beads with UV irradiation. Substituting DOM and turbidity, a constant quantity of humic acid (HA) and kaolin was dissolved in distilled water. This study was the unique application of pure PP beads and UV irradiation to investigate the influence of water backwashing period (BP) and time (BT) in the combined water treatment process of MF membrane. A combined module was composed of the MF membrane and the PP beads, those were fluidized between the gap of the alumina membrane and the acryl module case. The results of the combined process of the seven channels alumina MF (pore size: 0.4 μm) membrane and pure PP beads with water backwashing were compared with those of tubular alumina MF (pore size: 0.1 μm) membrane, to examine the effects of membrane shape and pore size on membrane fouling and treatment efficiency. The results were compared with our previous studies [24,25], which used tubular carbon fiber UF and TiO_2_ photocatalyst-coated PP [24], or pure PP beads [25] in the combined water treatment process.

## 2. Materials and Methods 

### 2.1. Membranes and Polypropylene (PP) Beads

The tubular alumina (NCMT-7231, pore size 0.1 μm) and seven channels alumina MF (HC04, pore size 0.4 μm) membranes, were manufactured in Nanopore Inc. (Seoul, Korea) and Dongseo Industry (Chungnam, Korea), respectively, were applied in this research. The specification of the tubular and seven channels alumina membranes was compared in Table 1. Cross-sectional area of the pore, flow velocity and Reynolds number (Re) in membrane channel were given for flow pattern. Turbulent flow (Re > 4000) was happened in NCMT-7231; however, laminar flow (Re < 2100) in HC10.

The pure polypropylene (PP) beads of 4–6 mm, which was purchased from SKC (Seoul, Korea), were employed in this research, those of the average weight was 39.9 mg. Substituting natural organic matters and fine inorganic particles in natural water sources, a constant quantity of HA and kaolin was dissolved in distilled water. In this experimental study, it was utilized as synthetic feed water. For photo-oxidation of DOM, two UV lamps (F8T5BLB, Sankyo, Tokyo, Japan) irradiated UV with 352 nm from the acryl module outside. 

### 2.2. Experimental Procedures

To eliminate the turbidity and DOM, the pure PP beads were fluidized in the space between the acryl module inside and the alumina membrane outside. Furthermore, a 100 mesh (0.150 mm) sieve, that was much smaller than 4–6 mm of the PP beads size employed here, was mounted at the outlet of the combined module, because of the PP beads loss out of the module. 

The combined water treatment process (6) of alumina MF membrane and the pure PP beads (7), which were used in the previous study [23], as displayed in Figure 1. A periodic water backwashing utilizing permeated water from the combined module was accomplished for the MF membrane. The combined module (6) was provided with the PP beads fluidizing between the gap of the MF membrane and the acryl module case. 

The three pressure gages (P) were installed at the membrane inlet, membrane outlet, and backwashing water inlet, to measure transmembrane pressure (TMP) and backwashing water pressure. Then, 10 L of the water, which was composed of HA and kaolin, was contained in the feed tank (1). To preserve a constant viscosity of water, the feedwater temperature was constantly maintained by a temperature control water circulator (3) (Model 1146, VWR, Atlanta, GA, USA). To maintain the homogeneous feed water, it was continuously mixed by a stirrer (4), and it flowed into the MF membrane inside by a pump (2) (Procon, Standex Co., Smyna, TN, USA). A flowmeter (5) (NP-127, Tokyo Keiso, Japan) measured the feed flow rate to the membrane module. Regulating valves (9) of both the bypass pipe of the pump (2) and the concentrate pipe could maintain constant flow rate and pressure of the feed water that flowed into the combined module. An electric balance (11) (Ohaus, Newark, NJ, USA) measured the permeate flux treated by both the MF membrane and the PP beads. During the permeate flux had not been measured, the permeate water flowed into the backwashing tank (13). To maintain a constant concentration of the feed water during operation, the treated water was recycled to the feed tank (1) after it was over a certain level in the water backwashing tank (13). Physical washing was accomplished by a brush inside the membrane channels after each of the three hours’ procedures. After that, the permeate flux was measured to evaluate the resistances of irreversible and reversible membrane fouling. The real pictures for the combined water treatment process and seven channel MF membranes were displayed in Figure 2.

To investigate the influence of the water backwashing period (BP), BP was changed from 2 to 10 min, and no backwashing (NBW), in the condition that fixed backwashing time (BT) at 10 s. Furthermore, to examine the BT influence, BT was changed from 30 to 6 s, and NBW, at stationary BP 10 min. HA and kaolin were set at 10 mg/L and 30 mg/L in all of the experimental research, respectively. During the total 180 min’s operation time, the permeate flux (J) was checked at each experimental condition. In the combined process of tubular NCMT-7231 MF, TMP was sustained constant at 1.8 bar, and the water backwashing pressure at 2.5 bar; however, in the case of seven channel HC04 MF, TMP was fixed at 0.8 bar, and the water backwashing pressure at 1.0 bar, because the HC04 had 4.49 times higher membrane surface than NCMT-7231. The feed flow rate was fixed at 1.0 L/min, and the feedwater temperature was at 20 °C in all experiments. The PP beads concentration was set at 40 g/L in the combined module.

### 2.3. Analytical Methods

The quality of feed and treated water was analyzed every 30 min during each experiment, for evaluating the treatment efficiencies of turbidity and DOM. Turbidity was checked by a turbidimeter (2100N, Hach, Ames, IA, USA). Before checking UV_254_ absorbance, each sample was filtered by a 0.2 μm syringe filter to eliminate turbidity. The UV_254_ absorbance was examined by a UV spectrophotometer (Genesys 10 UV, Thermo, Pittsburgh, PA, USA) to measure DOM. The detection limits of the turbidimeter and UV spectrophotometer were 0~4000 NTU (±0.001 NTU) and −0.1~3.0 cm^−1^ (±0.001 cm^−1^), respectively. 

### 2.4. Membrane Recovery Methods

After finishing each experiment, all of the synthetic solution was discharged from the combined water treatment system, and distilled water was circulated in the line of the system for 15 min. The PP beads were collected, and the ceramic membranes were separated from the module. Most of the fouling materials inside the alumina membrane could be removed by combusting at 550 °C in a furnace over 30 min. After cooling the membrane, it was immersed in a nitric acid (HNO_3_) of 15% for 24 h, and in a sodium hydroxide (NaOH) solution of 0.25 N for 3 h, to dissolve out organic or inorganic contaminants that remained inside the membrane. For rinsing and rejecting air in the membrane pore, it was kept in distilled water for 24 h. 

Before operating a new experiment, the water permeated flux (J_w_) was measured for evaluating the membrane recovery when a normal operation was performed with distilled water. If the error of J_w_ were less than 5%, the recovered membrane was installed inside the module for another experiment. The recovered membrane was applied in all of the experiments to minimize the impact of membrane condition on the treatment efficiency.

### 2.5. Resistance-in-Series Filtration Model

Applying the resistance-in-series filtration Equation (1) as the same method as the previous study [23], where ΔP is TMP, resistances of the membrane, boundary layer, and membrane fouling (R_m_, R_b_, R_f_) were evaluated from permeate flux (J) data.
J = ΔP/(R_m_ + R_b_ + R_f_)(1)

In detail, the equation was simplified to Equation (2) for a new membrane, because there were no resistances of the boundary layer and membrane fouling. Finally, the R_m_ could be extracted from J data for a new membrane.
J = ΔP/R_m_(2)

For the solution prepared with HA and kaolin, the equation was revised to Equation (3) at the initial time, and R_b_ could be calculated using initial J (J_0_) and R_m_ values.
J = ΔP/(R_m_ + R_b_)(3)

In addition, resistances of the irreversible and reversible membrane fouling (R_if_, R_rf_) could be determined from J values, before and after physical washing utilizing a brush inside the membrane.

## 3. Results and Discussions

The optimal water backwashing was investigated in the combined water treatment process of tubular (NCMT-7231) or seven channels (HC10) alumina MF membrane and pure PP beads with periodic water backwashing and UV irradiation. 

### 3.1. Influence of Water Backwashing Time (BT) on Membrane Fouling and Treatment Efficiency

The influence of water backwashing time was investigated for the solution of HA 10 mg/L, kaolin 30 mg/L. The resistances of membrane fouling (R_f_) maintained the highest values at NBW from 30 min’s operation, and the lowest at BT 20 s with BP 10 min until 120 min; however, the minimum at BT 30 s after 120 min, as compared in Figure 3a. It proved that the BT 20 s could be sufficient to inhibit the membrane fouling until 120 min’s operation; however, longer BT 30 s was more effective than BT 20 s after 120 min in this combined water treatment process of tubular alumina MF and PP beads. 

In our previous work [24], for the combined water treatment process of tubular carbon fiber ultrafiltration (UF) and photocatalyst-coated PP beads, the R_f_ maintained the lowest values at BT 30 s and the highest at NBW condition during all of 180 min’s operation. The previous result did not exactly agree with the trend of BT influence in this research using tubular alumina MF, because the UF membrane had a smaller pore size 0.05 μm than 0.1 μm of the alumina MF. It proved that the water backwashing time should be adjusted depending on membrane pore size in the combined water treatment process.

As presented in Figure 3b, the dimensionless permeate flux (J/J_0_), where J_0_ was the initial permeate flux predicted using the initial two data by an extrapolation method, as compared to investigate a BT influence on the relative decline of permeate flux. The J/J_0_ values overlapped almost at every BT condition; however, those at BT 20 s showed the highest from 10 to 30 min’s operation. In the previous work [24], for the combined process of tubular carbon fiber UF and photocatalyst-coated PP beads, the J/J_0_ tended to increase significantly as increasing BT from NBF to 30 s, because the longer water backwashing should more efficiently inhibit the cake accumulation on the membrane surface and the fouling inside the membrane.

As arranged in Table 2, the final J after 180 min’s operation (J_180_) increased dramatically from NBW to BT 30 s, because the final R_f_ (R_f,180_) decreased obviously as increasing BT. It proved that the permeate flux could maintain the highest values at the longest BT 30 s because the membrane fouling was inhibited effectively at BT 30 s in the combined water treatment process. In conclusion, the J_180_/J_0_ after 180 min’s operation at BT 30 s showed the highest 0.101, which was 1.28 times higher than 0.079 at NBW condition. However, the total treated water volume (V_T_) had the highest 3.13 L at BT 20 s, because J maintained higher from 10 to 30 min’s operation than those of other BT conditions, as shown in Figure 3b. Finally, the optimal BT condition could be 20 s in the combined water treatment process, because of the maximum V_T_ in our experimental BT conditions. In the previous work [24], for the combined process of tubular carbon fiber UF and photocatalyst-coated PP beads, the highest V_T_ 3.00 L could be acquired at BT 30 s, which was 1.33 times higher than the lowest 2.25 L at NBW. The optimal BT for the former result [24] could be 30 s otherwise, because of different pore sizes.

As compared in Table 2, the resistance of the boundary layer (R_b_), which was formed by concentration polarization on the membrane surface, was the lowest at BT 15 s in the combined process of tubular alumina MF and PP beads. It proved that the BT 15 s could reduce the concentration polarization on the membrane surface the most effectively. The R_f,180_ after 180 min’s operation at NBW condition showed the maximum 5.742 × 10^9^ kg/m^2^s, which was 1.25 times higher than the minimum 4.597 × 10^9^ kg/m^2^s at BT 30 s with BP 10 min.

Resistances of membrane, boundary layer, final, irreversible and reversible membrane fouling (R_m_, R_b_, R_f,180_, R_if_, R_rf_) were obviously compared as bar graphs in Figure 4. Because the R_rf_ had a lot of portions of R_f,180_, it could be a major membrane fouling in this combined water treatment process. It proved that most of the membrane fouling could be easily removed by physical cleaning. In addition, the R_rf_ decreased clearly from NBW to BT 30 s; however, the R_if_ did not show a constant trend depending on BT condition and was the lowest value at BT 10 s. It means that the BT 10 s could reduce the irreversible membrane fouling efficiently in our BT range.

As arranged in Table 3, the treatment efficiencies of turbidity showed almost constant values in the range of 99.5% and 99.7%, independent of the BT condition. It proved that the turbid matters could be removed effectively over 99.5% by microfiltration and PP beads adsorption in this combined water treatment process of tubular alumina MF and PP beads. In the previous work [24], for the combined process of tubular carbon fiber UF and photocatalyst-coated PP beads, the treatment efficiency of turbidity was maintained almost the same as 99.2–99.5%, independent of water BT. It agreed exactly with the result utilizing tubular alumina MF and pure PP beads.

The turbidity of real lake water located in Chuncheon, Korea, was the range of 33.3-39.2 NTU (average 36.3 NTU), as published by our group [27]. That of kaolin 30 mg/L solution in Table 3 was 144.5–162.5 NTU (average 153.5 NTU), which was 4.22 times higher than that of real lake water. Therefore, the kaolin solution used in this study could modify the severe quality of lake water.

As compared in Table 4, the treatment efficiency of UV_254_ absorbance, which could substitute the concentration of DOM (dissolved organic matters), did not present a constant trend, depending on BT; however, it showed the highest 92.4% at BT 20 s. It proved that the optimal BT condition could be 20 s for DOM treatment in this combined process of tubular alumina MF and PP beads. In the previous work [25], for the combined process of carbon fiber UF and photocatalyst-coated PP beads, the treatment efficiency of DOM showed the highest 67.3% at BT 30 s and increased dramatically as increasing BT from NBF to 30 s. It proved that the photocatalyst-coated PP beads were cleaned effectively by water backwashing as increasing BT from NBF to 30 s, and finally, the DOM could be photo-oxidized or adsorbed effectively by the PP beads and UV irradiation. The treatment efficiency of DOM was much lower than that using tubular alumina MF and pure PP beads in this research. It seems that both the membrane and beads could affect the DOM treatment. 

The UV_254_ absorbance of real lake water was in the range of 0.041–0.045 cm^−1^ (average 0.043 cm^−1^), as published by our group [27]. That of HA 10 mg/L solution in Table 4 was 0.246-0.345 cm^−1^ (average 0.300 cm^−1^), which was 7.00 times higher than that of real lake water. Therefore, the HA solution used in this research could modify the extreme condition of lake water quality.

### 3.2. Influence of Water Backwashing Period (BP) on Membrane Fouling and Treatment Efficiency

As compared in Figure 5a, the resistances of membrane fouling (R_f_) were maintained the highest and almost overlapped at BP 2 and 4 min condition during 180 min’s operation time. The R_f_ showed suddenly the lowest values at BP 6 min, which means the BP 6 min could be the optimal water backwashing period in this combined water treatment process of tubular alumina MF (NCMT-7231) and PP beads. However, the R_f_ increased dramatically at BP 8 min and decreased at BP 10 min, which proved that too longer BP more than 6 min could not be efficient to inhibit the membrane fouling. In addition, the R_f_ values increased a little at no water backwashing (NBW), which means too short or longer BP conditions could not reduce the membrane fouling in this combined water treatment.

As presented in Figure 5b, the dimensionless permeate flux (J/J_0_)*,* overlapped almost and maintained the lowest values at BP 2, 4, and 8 min. However, the J/J_0_ increased dramatically high at BP 6 min, because the membrane fouling was inhibited effectively by proper water backwashing period and permeate flux could maintain high values. Furthermore, it increased a little at BP 10 min, and abruptly high at NBW condition. This phenomenon could be explained by comparing the J_0_ as arranged in Table 5. Because the J_0_ values dropped suddenly at BP 10 min and NBW by longer BP condition, the J/J_0_ increased dramatically. 

As arranged in Table 5, the resistance of membrane (R_m_) was regulated at almost constant value by combustion at a furnace and washing by acid and alkali solution, as mentioned in Section 2. Materials and Methods. The final R_f_ (R_f,180_) value after 180 min’s operation at BP 2 min was 4.877 × 10^9^ kg/m^2^s, which was 1.24 times higher than 3.931 × 10^9^ kg/m^2^s at BP 6 min. Furthermore, the total treated volume (V_T_) showed the highest 3.10 L at BP 6 min, which was 1.57 times higher than 2.56 L at BP 2 min. In conclusion, the optimal BP condition could be 6 min, because of the minimum membrane fouling and the maximum total treated volume in this combined water treatment process of tubular alumina MF (NCMT-7231) and PP beads.

In bar graphs of Figure 6, resistances of membrane, boundary layer, final, irreversible and reversible membrane fouling (R_m_, R_b_, R_f,180_, R_if_, R_rf_) were compared clearly. The R_rf_ was a major membrane fouling in this combined water treatment process. It proved that most of the membrane fouling could be easily removed by physical cleaning at the BP influence experiment, which matched with the result of BT influence. In addition, the R_b_ decreased dramatically as decreasing BP from NBW to BP 2 min. It means that the more frequent water backwashing could weaken the boundary layer of concentration polarization on the membrane surface excellently.

As arranged in Table 6, the treatment efficiency of turbidity did not have a constant trend, depending on BP; however, the highest efficiency was 99.7% at BP 4 and 6 min. Furthermore, the lowest value showed 98.4% at NBW, but the difference between the highest and lowest efficiency was just 1.3%. It proved that the turbidity could be powerfully treated over 98.4% by microfiltration and PP beads adsorption, independent of water BP condition in the combined process of tubular alumina MF and PP beads. In our previous work [25], for the combined process of tubular carbon fiber UF and pure PP beads, the turbidity treatment efficiency was a little higher 99.7% than 98.4% of this research, because MF had a larger pore size than UF.

As compared in Table 7, the treatment efficiency of UV_254_ absorbance (DOM) did not present a constant trend like that of turbidity. The maximal efficiency showed 94.3% at BP 8 min, and the lowest value was 91.6% at BP 4 min; however, the gap of the two values was 2.7%, which was a little higher than that of turbidity. It means that DOM could be excellently rejected over 91.6% by PP beads adsorption and UV photo-oxidation in the combined water treatment process. The DOM treatment efficiencies of PP beads adsorption and UV photo-oxidation were 5.4% and 0.7% respectively as published by our group in Desal. Water Treat. [25]. Application of photocatalysis contributed to the improvement of the permeate flux compared with photolysis by 16–35% using a ceramic membrane with ZrO_2_ separation layer [11]. However, there was no data for the effect of photocatalysis on treatment efficiency in this research.

### 3.3. Influence of Membrane Shape (Tubular or Seven Channel) on Membrane Fouling and Treatment Efficiency

To investigate the influence of membrane shape, the resistance of membrane fouling (R_f_) was compared depending on the water backwashing period (BP) in the combined process of seven channel alumina MF (HC04) and pure PP beads, as presented in Figure 7a. The R_f_ values maintained the lowest at BP 2 min and the highest at BP 10 min during all of 180 min’s operation. Those showed a dramatic trend to increase as increasing BP from 2 to 10 min; however, those decreased a little at NBW condition. It means that water BP 10 min could not reduce the membrane fouling in this combined process. Finally, the BP 2 min could be the optimal water backwashing period in the combined process of seven channel alumina MF and PP beads. It was an obviously different trend, comparing that the R_f_ at BP 6 min was lowest in this combined process of tubular alumina MF (NCMT-7231) and PP beads, because of different membrane shape, as shown in Figure 5a.

As presented in Figure 7b to investigate the BP influence on relative permeate flux, the J/J_0_ values showed a decreasing trend dramatically through all of the operation time, as increasing BP from 2 to 10 min; however, those at NBW increased a little higher than those of BP 10 min. It proved that the BP of 10 min could be a too long period to inhibit the membrane fouling in this combined process using seven channel MF. As summarized in Table 8, the final J_180_/J_0_ after 180 min’s operation at BP 2 min showed the highest 0.422, which was 2.14 times higher than the lowest 0.197 at BP 8 min. It proved that the highest permeate flux could be acquired at BP 2, which was the most frequent water backwashing condition because the membrane fouling could be inhibited by frequent backwashing. In conclusion, the maximum V_T_ of 5.79 L could be acquired at BP 2 min, because the permeate flux could sustain highly through all of 180 min’s operation. The J/J_0_ overlapped almost and maintained the lowest values at BP 2, 4, and 8 min in the combined process using tubular MF, as presented in Figure 5b. However, the J/J_0_ increased dramatically high at BP 6 min. This dissimilar trend was happened, because the seven channel MF membrane could have a different membrane fouling mechanism, compared with tubular MF.

As presented in Table 8, the final R_f_ (R_f,180_) showed the highest 7.287×10^9^ kg/m^2^s at BP 8 min, which was 3.11 times higher than the lowest 2.342×10^9^ kg/m^2^s at BP 2 min. It means that a longer BP than 8 min could not inhibit the membrane fouling by water backwashing in the combined process using tubular MF. However, in the combined process using tubular MF, the R_f,180_ value at BP 2 min was the maximum, and that of BP 6 min was the minimum, as arranged in Section 3.2, because of different membrane shapes. It proved that the water backwashing period should be adjusted, depending on the membrane shape such as tubular or multi-channel. In addition, the maximal 7.287×10^9^ kg/m^2^s of R_f,180_ using seven channel MF was much higher than 4.849×10^9^ kg/m^2^s of tubular MF; however, the minimal 2.342×10^9^ kg/m^2^s of seven channel MF was lower than 3.931×10^9^ kg/m^2^s of tubular MF. It proved that water backwashing could inhibit the membrane fouling better for seven channel membranes than a tubular membrane.

In the bar graphs in Figure 8, all of the resistances were compared to investigate each portion of total resistance. The R_rf_ showed a major resistance of total membrane fouling, and the R_if_ was a minor one, similar to the results of BT and BP for combined process using tubular MF. Moreover, the R_f,180_ and R_rf_ decreased dramatically as decreasing BP from 8 to 2 min; however, the R_if_ showed a decreasing trend obviously, as decreasing BP from NBW to 2 min. It means that the more frequent water backwashing could be more effective to control the membrane fouling, especially the irreversible fouling, for seven channel membrane than tubular membrane. Furthermore, the R_b_ did not show a constant trend, depending on BP; however, it decreased dramatically as shorter BP condition from NBW to BP 2 min in the combined process using tubular MF, because of different membrane shapes.

As summarized in Table 9, the treatment efficiency of turbidity showed almost constant in the range of 99.0% and 99.1%. In the combined process using tubular MF, the highest efficiency was 99.7% at BP 4 and 6 min, and the lowest 98.4% at NBW, as arranged in Table 6. The highest 99.7% for tubular MF was a little higher than 99.1% of seven channel MF because the tubular MF had a smaller pore size 0.1 μm than 0.4 μm of seven channel MF.

As compared in Table 10, the treatment efficiency of DOM did not show a constant trend, depending on BP. The highest efficiency showed 82.4% at BP 6 min, and the lowest was 80.1%; however, the difference between the two values was only 2.3%. It proved that the DOM treatment was not affected by the water backwashing period in the combined process using seven channel MF. For the process using tubular MF as shown in Table 7, the DOM treatment efficiency did not present a constant trend like this result. The maximal efficiency showed 94.3% at BP 8 min, and the lowest value was 91.6% at BP 4 min, which was higher than that of seven channel MF. It means that the tubular shape membrane could be more excellent to treat DOM than seven channel membrane.

## 4. Conclusions

In this research, the optimal water backwashing was investigated in the combined water treatment process of alumina MF and pure PP beads. Additionally, the influence of membrane shape (tubular of seven channel) was examined, depending on the water backwashing period. The results of water backwashing were compared with those of the previous study [24,25] in the combined process of the tubular carbon fiber UF and photocatalyst-coated PP beads or pure PP beads. In conclusion, the following results could be found out from these investigations.
(1)The dimensionless permeate flux (J_180_/J_0_) after 180 min’s operation at BT 30 s showed the highest value. However, the total treated water volume (V_T_) had the highest at BT 20 s, because J maintained higher from 10 to 30 min’s operation than those of other BT conditions. Finally, the optimal BT condition could be 20 s in the combined water treatment process, because of the maximum V_T_ in our experimental BT conditions. In the previous work [24], for the combined process of tubular carbon fiber UF and photocatalyst-coated PP beads, the maximum V_T_ could be acquired at BT 30 s, which agreed with this result of the combined process of tubular alumina MF and pure PP beads.(2)The final R_f_ (R_f,180_) value after 180 min’s operation at BP 2 min was the highest at BP 6 min. Furthermore, the V_T_ showed the highest 3.10 L at BP 6 min. In conclusion, the optimal BP condition could be 6 min, because of the minimum membrane fouling and the maximum total treated volume in this combined water treatment process of tubular alumina MF and PP beads.(3)The resistance of reversible membrane fouling (R_rf_) showed a major resistance of total membrane fouling, and that of irreversible membrane fouling (R_if_) was a minor one, in the combined process using tubular or seven channel MF. The R_if_ showed a decreasing trend obviously, as decreasing BP from NBW to 2 min for seven channel MF. It means that the more frequent water backwashing could be more effective to control the membrane fouling, especially irreversible fouling, for seven channel membranes than tubular membranes.

## Figures and Tables

**Figure 1 membranes-12-00092-f001:**
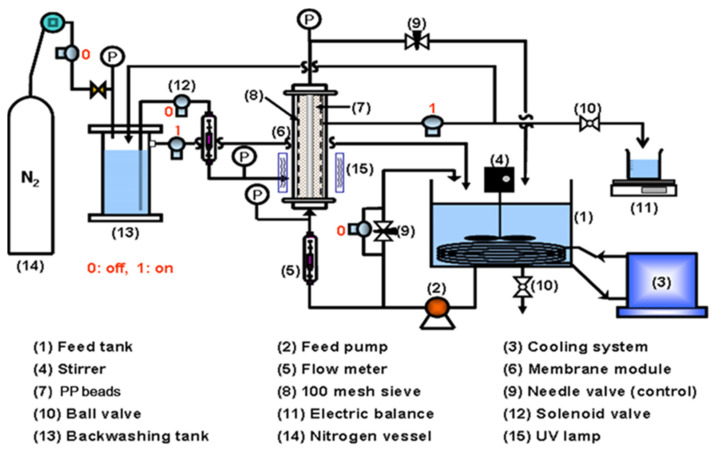
Apparatus of combined water treatment process of tubular or seven channel MF membrane and PP beads with periodic water backwashing.

**Figure 2 membranes-12-00092-f002:**
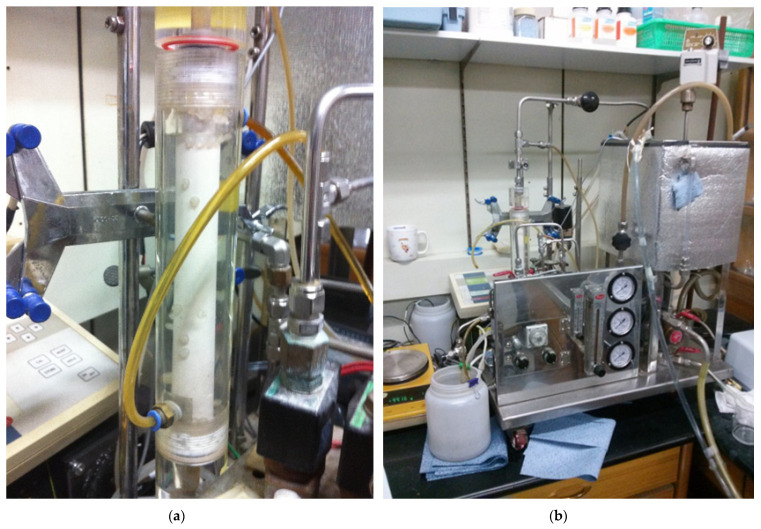
Combined water treatment process of tubular or seven channel MF membrane and PP beads with periodic water backwashing: (**a**) seven channel MF membrane; (**b**) total combined system.

**Figure 3 membranes-12-00092-f003:**
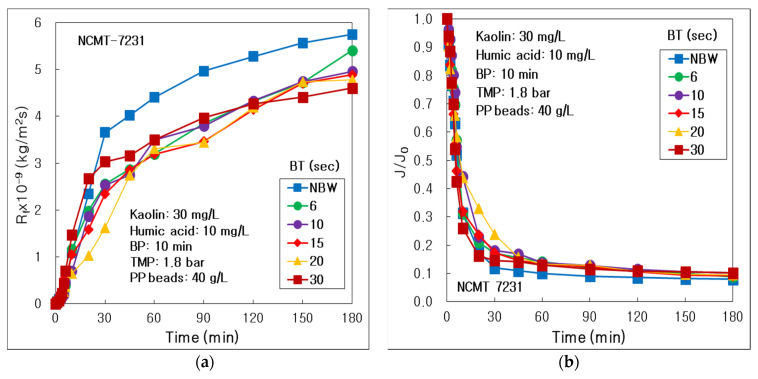
Influence of water backwashing time (BT) in the combined process of tubular alumina MF (NCMT-7231) and PP beads with UV irradiation and periodic water backwashing: (**a**) Resistance of membrane fouling; (**b**) Dimensionless permeate flux.

**Figure 4 membranes-12-00092-f004:**
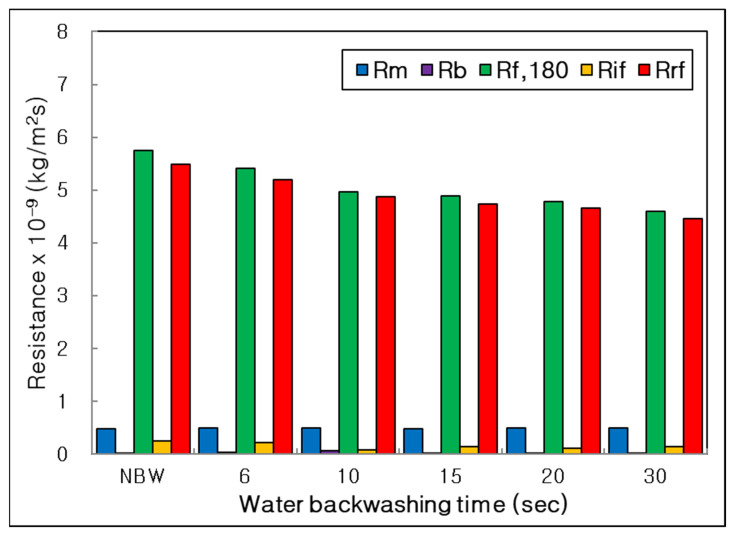
Influence of water backwashing time on resistances of membrane, boundary layer, final, irreversible and reversible membrane fouling (R_m_, R_b_, R_f,180_, R_if_, R_rf_) in the combined process of tubular alumina MF (NCMT-7231) and PP beads with UV irradiation and periodic water backwashing.

**Figure 5 membranes-12-00092-f005:**
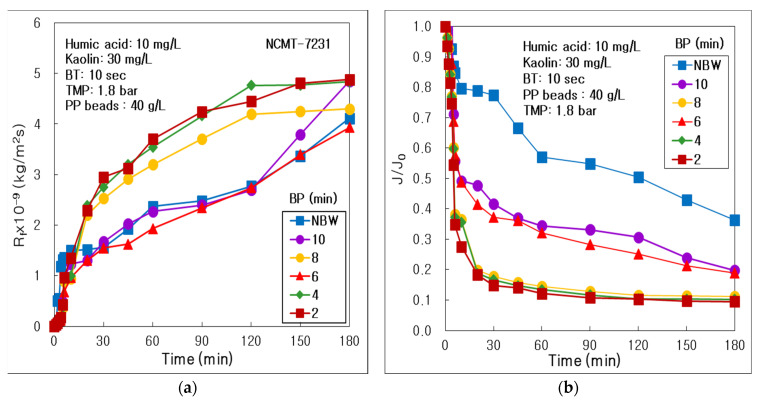
Influence of water backwashing period (BP) in the combined process of tubular alumina MF (NCMT-7231) and PP beads with UV irradiation and periodic water backwashing: (**a**) Resistance of membrane fouling; (**b**) Dimensionless permeate flux.

**Figure 6 membranes-12-00092-f006:**
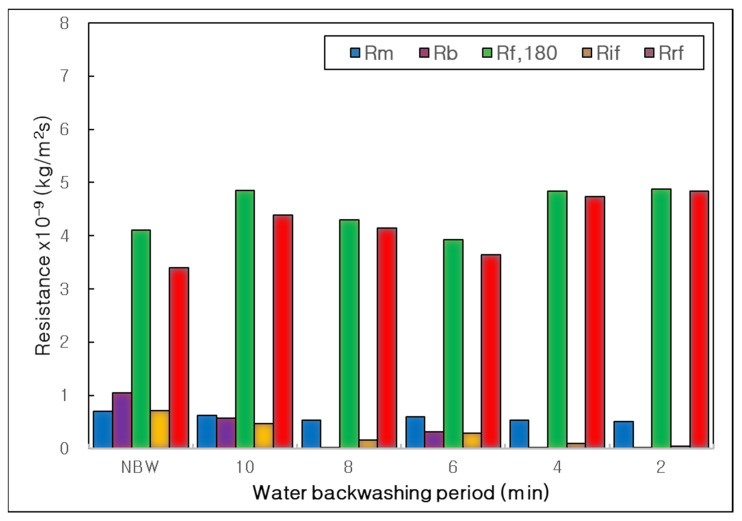
Influence of water backwashing period on resistances of membrane, boundary layer, final, irreversible and reversible membrane fouling (R_m_, R_b_, R_f,180_, R_if_, R_rf_) in the combined process of tubular alumina MF (NCMT-7231) and PP beads with UV irradiation and periodic water backwashing.

**Figure 7 membranes-12-00092-f007:**
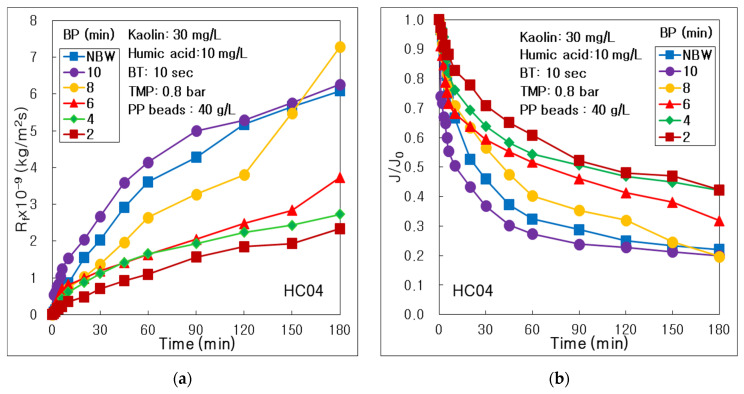
Influence of water backwashing period (BP) in the combined process of seven channel alumina MF (HC04) and PP beads with UV irradiation and periodic water backwashing: (**a**) Resistance of membrane fouling; (**b**) Dimensionless permeate flux.

**Figure 8 membranes-12-00092-f008:**
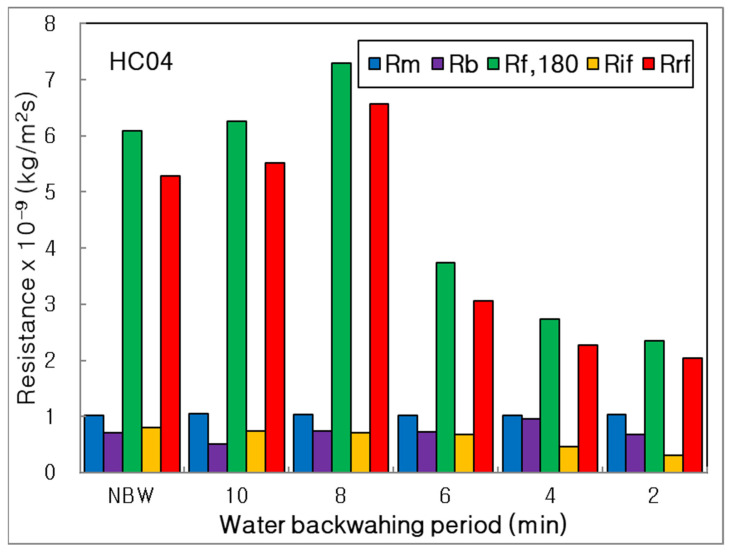
Influence of water backwashing period on resistances of membrane, boundary layer, final, irreversible and reversible membrane fouling (R_m_, R_b_, R_f,180_, R_if_, R_rf_) in the combined process of seven channel alumina MF (HC04) and PP beads with UV irradiation and periodic water backwashing.

**Table 1 membranes-12-00092-t001:** Specification of tubular and seven channel alumina ceramic MF membranes employed in this experimental research.

Membrane Model	NCMT-7231	HC10
Pore size (μm)	0.4	0.1
No. of channels	1	7
Outer diameter (mm)	8	20
Inner diameter (mm)	6	4
Length (mm)	254	245
Surface area (cm^2^)	47.9	215
Cross sectional area of pore (cm^2^)	0.283	0.880
Flow velocity in channel (cm/s)	58.9	18.9
Reynolds number in channel	3510	751
Material	α-alumina	α-alumina
Company	Nanopore Inc. (Korea)	Dongseo Industry (Korea)

**Table 2 membranes-12-00092-t002:** Influence of water backwashing time on filtration factors in the combined process of tubular alumina MF (NCMT-7231) membrane and PP beads with periodic water backwashing.

BT (s)	NBW *	6	10	15	20	30
R_m_ × 10^−9^ (kg/m^2^s)	0.484	0.490	0.489	0.484	0.493	0.495
R_b_ × 10^−9^ (kg/m^2^s)	0.011	0.037	0.070	0.007	0.013	0.023
R_f,180_ × 10^−9^ (kg/m^2^s)	5.742	5.404	4.958	4.885	4.781	4.597
R_if_ × 10^−9^ (kg/m^2^s)	0.247	0.214	0.086	0.147	0.117	0.136
R_rf_ × 10^−9^ (kg/m^2^s)	5.495	5.190	4.873	4.739	4.663	4.461
J_0_ (L/m^2^hr)	1282	1207	1137	1294	1255	1225
J_180_ (L/m^2^hr)	102	107	115	118	120	124
J_180_/J_0_	0.079	0.089	0.101	0.091	0.096	0.101
V_T_ (L)	2.43	2.76	2.80	2.88	3.13	2.60

* no backwashing.

**Table 3 membranes-12-00092-t003:** Water quality and treatment efficiency of turbidity in the combined process of tubular alumina MF (NCMT-7231) and PP beads for the influence of water backwashing time.

BT (s)	Turbidity (NTU)				Average Treatment Efficiency (%)
	Feed Water		Treated Water		
	Range	Average	Range	Average	
NBW *	156.0~157.0	156.8	0.447~0.512	0.488	99.7
6	153.0~155.0	154.2	0.668~0.722	0.694	99.6
10	153.0~155.0	154.0	0.672~0.725	0.694	99.5
15	143.0~145.0	144.5	0.607~0.620	0.613	99.6
20	147.0~149.0	148.0	0.611~0.657	0.635	99.6
30	162.0~163.0	162.5	0.612~0.684	0.635	99.6

* no backwashing.

**Table 4 membranes-12-00092-t004:** Water quality and treatment efficiency of DOM (UV_254_ absorbance) in the combined process of tubular alumina MF (NCMT-7231) and PP beads for the influence of water backwashing time.

BT (s)	UV_254_ Absorbance (cm^−1^)				Average Treatment Efficiency (%)
	Feed Water		Treated Water		
	Range	Average	Range	Average	
NBW *	0.322~0.324	0.323	0.038~0.044	0.042	87.1
6	0.311~0.313	0.313	0.031~0.038	0.035	89.0
10	0.317~0.325	0.263	0.037~0.046	0.041	87.3
15	0.238~0.303	0.246	0.040~0.045	0.042	84.2
20	0.249~0.331	0.264	0.016~0.023	0.020	92.4
30	0.341~0.349	0.345	0.041~0.054	0.045	86.9

* no backwashing.

**Table 5 membranes-12-00092-t005:** Influence of water backwashing period on filtration factors in the combined process of tubular alumina MF (NCMT-7231) membrane and PP beads with periodic water backwashing.

BP (min)	NBW *	10	8	6	4	2
R_m_ × 10^−9^ (kg/m^2^s)	0.694	0.626	0.528	0.602	0.534	0.508
R_b_ × 10^−9^ (kg/m^2^s)	1.050	0.566	0.017	0.316	0.017	0.005
R_f,180_ × 10^−9^ (kg/m^2^s)	4.112	4.849	4.299	3.931	4.834	4.877
R_if_ × 10^−9^ (kg/m^2^s)	0.719	0.465	0.156	0.283	0.098	0.040
R_rf_ × 10^−9^ (kg/m^2^s)	3.393	4.384	4.143	3.648	4.736	4.837
J_0_ (L/m^2^hr)	364	533	1166	692	1152	1237
J_180_ (L/m^2^hr)	132	105	131	131	118	118
J_180_/J_0_	0.363	0.197	0.112	0.189	0.102	0.095
V_T_ (L)	3.01	2.63	2.72	3.10	2.56	2.56

* no backwashing.

**Table 6 membranes-12-00092-t006:** Water quality and treatment efficiency of turbidity in the combined process of tubular alumina MF (NCMT-7231) and PP beads for the influence of water backwashing period.

BP (min)	Turbidity (NTU)				Average Treatment Efficiency (%)
	Feed Water		Treated Water		
	Range	Average	Range	Average	
NBW *	41.1~43.5	42.2	0.623~0.723	0.687	98.4
10	40.3~43.6	41.7	0.498~0.612	0.547	98.7
8	47.4~49.2	48.1	0.202~0.262	0.299	99.5
6	45.8~61.3	52.6	0.091~0.313	0.176	99.7
4	47.2~51.5	49.6	0.136~0.239	0.173	99.7
2	43.6~51.9	47.9	0.523~0.832	0.694	98.6

* no backwashing.

**Table 7 membranes-12-00092-t007:** Water quality and treatment efficiency of DOM (UV_254_ absorbance) in the combined process of tubular alumina MF (NCMT-7231) and PP beads for the influence of water backwashing period.

BP (min)	UV_254_ Absorbance (cm^−1^)				Average Treatment Efficiency (%)
	Feed Water		Treated Water		
	Range	Average	Range	Average	
NBW *	0.226~0.236	0.234	0.017~0.019	0.018	92.4
10	0.221~0.249	0.236	0.012~0.018	0.015	93.7
8	0.021~0.304	0.299	0.011~0.022	0.017	94.3
6	0.247~0.351	0.315	0.013~0.036	0.021	93.4
4	0.211~0.341	0.289	0.021~0.029	0.024	91.6
2	0.288~0.315	0.297	0.009~0.038	0.018	93.8

* no backwashing.

**Table 8 membranes-12-00092-t008:** Influence of water backwashing period on filtration factors in the combined process of seven channel alumina MF (HC04) membrane and PP beads with periodic water backwashing.

BP (min)	NBW *	10	8	6	4	2
R_m_ × 10^−9^ (kg/m^2^s)	1.019	1.052	1.041	1.025	1.019	1.030
R_b_ × 10^−9^ (kg/m^2^s)	0.710	0.511	0.748	0.720	0.962	0.682
R_f,180_ × 10^−9^ (kg/m^2^s)	6.086	6.264	7.287	3.733	2.732	2.342
R_if_ × 10^−9^ (kg/m^2^s)	0.797	0.741	0.717	0.673	0.456	0.309
R_rf_ × 10^−9^ (kg/m^2^s)	5.288	5.523	6.571	3.060	2.276	2.033
J_0_ (L/m^2^hr)	163	181	158	162	142	165
J_180_ (L/m^2^hr)	36	36	31	52	60	70
J_180_/J_0_	0.221	0.200	0.197	0.319	0.420	0.422
V_T_ (L)	3.60	3.28	3.98	4.95	4.83	5.79

* no backwashing.

**Table 9 membranes-12-00092-t009:** Water quality and treatment efficiency of turbidity in the combined process of seven channel alumina MF (HC04) and PP beads for the influence of water backwashing period.

BP (min)	Turbidity (NTU)				Average Treatment Efficiency (%)
	Feed Water		Treated Water		
	Range	Average	Range	Average	
NBW *	56.9~61.5	59.3	0.506~0.637	0.567	99.1
10	56.5~61.1	58.5	0.482~0.658	0.557	99.1
8	67.2~72.8	69.8	0.674~0.867	0.752	99.1
6	72.8~75.1	73.9	0.601~0.735	0.677	99.1
4	68.2~71.8	70.0	0.574~0.689	0.647	99.1
2	76.4~78.2	77.2	0.632~0.963	0.805	99.0

* no backwashing.

**Table 10 membranes-12-00092-t010:** Water quality and treatment efficiency of DOM (UV_254_ absorbance) in the combined process of seven channel alumina MF (HC04) and PP beads for the influence of water backwashing period.

BP (min)	UV_254_ Absorbance (cm^−1^)				Average Treatment Efficiency (%)
	Feed Water		Treated Water		
	Range	Average	Range	Average	
NBW *	0.524~0.631	0.578	0.097~0.127	0.111	80.8
10	0.441~0.513	0.482	0.081~0.114	0.096	80.2
8	0.583~0.608	0.598	0.108~0.120	0.114	80.9
6	0.584~0.624	0.609	0.089~0.121	0.107	82.4
4	0.541~0.609	0.577	0.098~0.119	0.108	81.2
2	0.524~0.584	0.560	0.101~0.122	0.111	80.1

* no backwashing.

## Data Availability

Not applicable.

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
