# Peer review of "Optimal Water Backwashing Condition in Combined Water Treatment of Alumina Microfiltration and PP Beads"

_membranes, 2022, doi:10.3390/membranes12010092_

Round 1

Reviewer 1 Report

Presented manuscript entitled “Strategy of water backwashing in combined water treatment of alumina microfiltration and PP beads” is, in my opinion, written with very chaotic English. There are a lot of poorly constructed sentences and mistakes throughout the text which need to be corrected, e.g. “[…] membrane fouling was happened by […]”, “[…] to overcome the restrictions of only a method.”, “To inspect the influence […]” etc.

Below, please find my detailed comments and questions regarding the manuscript:

  • Introduction is the weakest point of presented article. Each paragraph just slides on the surfaces of the topics and they do not form a whole. Authors must rebuilt it, with major English corrections.
  • In the line 32 Authors mention NOM (natural organic matter), while in line 86 “DOM” appears, without a shortcut explanation. Is it the same? Please explain.
  • PP breads shortcut is explained in line 81, while it first appears already in line 56.
  • Lines 79-84: I understand that presented article is a continuation of previous research [22-24] however, it is better to describe mentioned research, pointing differences, not only give reference.
  • For the first time I heard about “turbid matters”. Is it not just “turbidity”?
  • Figure 1: What is the meaning of “P” in the circle on the scheme?
  • Lines 157-162: “UV254 absorbance was examined by a UV spectrophotometer for measuring turbid matters and DOM”. However later: “Before checking UV254 absorbance, each sample was filtered by 0.2 µm syringe filter to eliminate turbid matters”. One sentence contradicts the other.
  • Chapters 3-4 (Conclusions are wrongly numbered as chapter 5) are written much better than Introduction and Materials and Methods. In this part I did not find major English and factual errors. However, still, chapter 3 presents only obtained results, with almost no discussion. Only at the beginning Authors tried to compare obtained results with their previous research. There is no comparison with other literature (Reference list contains only 24 positions). I suggest expanding this part.

Concluding, I suggest a major revision of presented manuscript.

Author Response

First of all, thanks a lot for your kind and detail comments. Answering each comment and revising the article were given as an attached file.

Reviewer 2 Report

Please provide real pictures for the apparatus setting. Also the membranes.

Please state the novelty of your research, I dont see any new knowledge presented?

Have you done experiment with real water?

Present some data including multiple backwash, if possible?

What is the impact of the beads on the life time of the membrane?

 Impact of the flow rate on the backwash time and frequency?

Author Response

(The authors gave the same response as above.)

Reviewer 3 Report

Fouling is a typical problem of membrane separation process. Development of method for fouling mitigation and for modelling of fouling layer build up has high relevance for the practice. There can be found numerous paper focused on the membrane fouling. Authors deal with water backwashing for minimized fouling investigating the effect of backwashing time and backwashing cycle, and analyse the change of membrane resistances. Therefore, the topic of the manuscript can be considered as interesting for the readers. The manuscript is generally logically and well structured. Introduction section summarizes well the background of the research and research motivations. The key words are selected well. Applied methods are adequate to the sample characteristics. Materials and methods (experimental setup, experiments) are given clearly but not in details. The manuscript contains interesting results but results are not discussed in details with references.

Comments, suggestions:

Please cite as ‘usual form’ of your previous papers (see line 81-84)

Authors compared the fouling in two different type membranes (diameter, surface are) with different pore size. Please give information related to flow velocity (feed, permeate), flow patterns etc. in the tubular and seven channel membranes.

Please give subsection in materials and method section (experimental set-up, analytical methods..etc.)

Measurements to determine the parameters of resistance-in-series model should be mentioned in Materials and methods section.

How was the concentration of HA and kaolin determined? Fouling is influenced by matrix effects, as well. Which tape of wastewater these concentration ranges can model?

The visibility of Figure 2-3 is ow (mainly axis titles, labels etc). Please improve it.

The frame of Figure 4 and 7 is partly missing. Please unify the figure style.

X axis title of figure 7: ‘backwashing’

Results are not discussed with references. Authors cited their own previous studies.

Author Response

(The authors gave the same response as above.)

Round 2

Reviewer 1 Report

Authors answered my questions and corrected the manuscript. In this regard I suggest acceptance of presented article in the revised form.

Reviewer 3 Report

The manuscript has an interesting topic that can provide interesting information for the readers. Authors have revised the manuscript thoroughly according to reviewers’ comments and suggestions. After the revision the overall scientific quality of the MS has been improved. Amendment of Introduction section, further and more detailed information added to Materials and methods section (and more structured methodology section), and more detailed discussion of experimental results made the manuscript more complete and clear. I accept all answers and modifications made by the authors.